# Large and fast human pyramidal neurons associate with intelligence

**Natalia A Goriounova[1]\*, Djai B Heyer[1†], René Wilbers[1†], Matthijs B Verhoog[1], Michele Giugliano[2,3,4], Christophe Verbist[2], Joshua Obermayer[1], Amber Kerkhofs[1], Harriët Smeding[5], Maaike Verberne[5], Sander Idema[6], Johannes C Baayen[6], Anton W Pieneman[1], Christiaan PJ de Kock[1], Martin Klein[7], Huibert D Mansvelder[1]\***

[1]Department of Integrative Neurophysiology, Amsterdam Neuroscience, Center for Neurogenomics and Cognitive Research (CNCR), Vrije Universiteit Amsterdam, Amsterdam, The Netherlands; [2]Department of Biomedical Sciences, University of Antwerp, Antwerp, Belgium; [3]Department of Computer Science, University of Sheffield, Sheffield, United Kingdom; [4]Brain Mind Institute, Lausanne, Switzerland; [5]Department of Psychology, Stichting Epilepsie Instellingen Nederland (SEIN), Zwolle, The Netherlands; [6]Department of Neurosurgery, VU medical center (VUmc), Amsterdam, The Netherlands; [7]Department of Medical Psychology, VU medical center (VUmc), Amsterdam, The Netherlands

**\*For correspondence:**
n.a.goriounova@vu.nl (NAG);
h.d.mansvelder@vu.nl (HDM)

[†]These authors contributed equally to this work

**Competing interests:** The authors declare that no competing interests exist.

**Abstract** It is generally assumed that human intelligence relies on efficient processing by neurons in our brain. Although grey matter thickness and activity of temporal and frontal cortical areas correlate with IQ scores, no direct evidence exists that links structural and physiological properties of neurons to human intelligence. Here, we find that high IQ scores and large temporal cortical thickness associate with larger, more complex dendrites of human pyramidal neurons. We show in silico that larger dendritic trees enable pyramidal neurons to track activity of synaptic inputs with higher temporal precision, due to fast action potential kinetics. Indeed, we find that human pyramidal neurons of individuals with higher IQ scores sustain fast action potential kinetics during repeated firing. These findings provide the first evidence that human intelligence is associated with neuronal complexity, action potential kinetics and efficient information transfer from inputs to output within cortical neurons.
DOI: https://doi.org/10.7554/eLife.41714.001

## Introduction

A fundamental question in neuroscience is what properties of neurons lie at the heart of human intelligence and underlie individual differences in mental ability. Thus far, experimental research on the neurobiological basis of intelligence has largely ignored the neuronal level and has not directly tested what role human neurons play in cognitive ability, mainly due to the inaccessibility of human neurons. Instead, research has either been focused on finding genetic loci that can explain part of the variance in intelligence (Spearman's *g*) in large cohorts (*Lam et al., 2017*; *Sniekers, 2017*; *Trampush et al., 2017*; *Coleman et al., 2018*) or on identifying brain regions of which structure or function correlate with IQ scores (*Karama et al., 2009*; *Hulshoff Pol et al., 2006*; *Narr et al., 2007*; *McDaniel, 2005*; *Deary et al., 2010*). Some studies have highlighted that variability in brain volume and intelligence may share a common genetic origin (*Hulshoff Pol et al., 2006*; *Posthuma et al., 2002*; *Sniekers, 2017*), and individual genes that were identified as

**eLife digest** Our brains are made up of almost 100 billion brain cells. Each of them acts like a small chip: they collect, process and pass on information in the form of electrical signals. In brain areas that integrate different types of information, such as frontal and temporal lobes, brain cells have larger dendrites – long projections specialized to collect signals. Theoretical studies predict that larger dendrites help cells to initiate electrical signals faster.

Because of difficulty in accessing human neurons, it has been unknown whether any of these features also relate to human intelligence. Previous studies have revealed that people with a higher IQ have a thicker outer layer (the cortex) in areas such as the frontal and temporal lobes. But does a thicker cortex also contain cells with larger dendrites and is their role different?

To test whether smarter brains are equipped with faster and larger cells, Goriounova et al. studied 46 people who needed surgery for brain tumors or epilepsy. Each took an IQ test before the operation. To access the diseased tissue deep in the brain, the surgeon also removed small, undamaged samples of temporal lobe. These samples still contained living cells and their electrical signals were measured in the lab. The experiments showed that cells from people with a higher IQ had larger dendrites that transported information more quickly, especially when they are very active. Computer models were then used to understand how these findings can lead to more efficient information transfer in human neurons.

Traditionally, research on human intelligence has focused on three main strategies: to study brain structure and function, to find genes associated with intelligence and to study the connection between our mind and behavior. Goriounova et al. are the first to take the single-cell perspective and link cell properties to human intelligence. The findings could help connect these separate approaches, and explain how genes for intelligence lead to thicker cortices and faster reaction times in people with higher IQ.

DOI: https://doi.org/10.7554/eLife.41714.002

associated with IQ scores might aid intelligence by facilitating neuron growth (*Sniekers, 2017*; *Coleman et al., 2018*) and directly influencing neuronal firing (*Lam et al., 2017*).

Intelligence is a distributed function that depends on activity of multiple brain regions (*Deary et al., 2010*). Structural and functional magnetic resonance imaging studies in hundreds of healthy subjects revealed that cortical volume and function of specific areas correlate with *g* (*Karama et al., 2009*; *Choi et al., 2008*; *Narr et al., 2007*). In particular, areas located in the frontal and temporal cortices show multiple correlations of grey matter thickness and functional activation with IQ scores: individuals with high IQ show larger grey matter volume of, for instance, Brodmann areas 21 and 38 (*Choi et al., 2008*; *Deary et al., 2010*; *Karama et al., 2009*; *Narr et al., 2007*). Cortical grey matter consists for a substantial part of dendrites (*Chklovskii et al., 2002*; *Ikari and Hayashi, 1981*), which receive and integrate synaptic information and strongly affect functional properties of neurons (*Bekkers and Häusser, 2007*; *Eyal et al., 2014*; *Vetter et al., 2001*). Especially higher order association areas in temporal and frontal lobes in humans harbor pyramidal neurons of extraordinary dendritic size and complexity (*Elston, 2003*; *Mohan et al., 2015*) that may constitute variation in cortical thickness, neuronal function, and ultimately IQ. These neurons and their connections form the principal building blocks for coding, processing, and information storage in the brain and give rise to cognition (*Salinas and Sejnowski, 2001*). Given their vast number in the human neocortex, even the slightest change in efficiency of information transfer by neurons may translate into large differences in mental ability. However, whether and how the activity and dendritic structure of single human neurons support human intelligence has not been tested.

To investigate whether structural and functional properties of neurons of the human temporal cortex associate with general intelligence, we collected a unique multimodal data set from 46 human subjects containing single cell physiology (31 subjects, 129 neurons), neuronal morphology (25 subjects, 72 neurons), pre-surgical MRI scans and IQ test scores (35 subjects, *Figure 1*, data available at the Dryad Digital Repository: https://doi.org/10.5061/dryad.83dv5j7).

Human cortical brain tissue was removed as a part of surgical treatment for epilepsy or tumor (*Table 1*). The tissue almost exclusively originated from middle temporal gyrus, approximately 4 cm

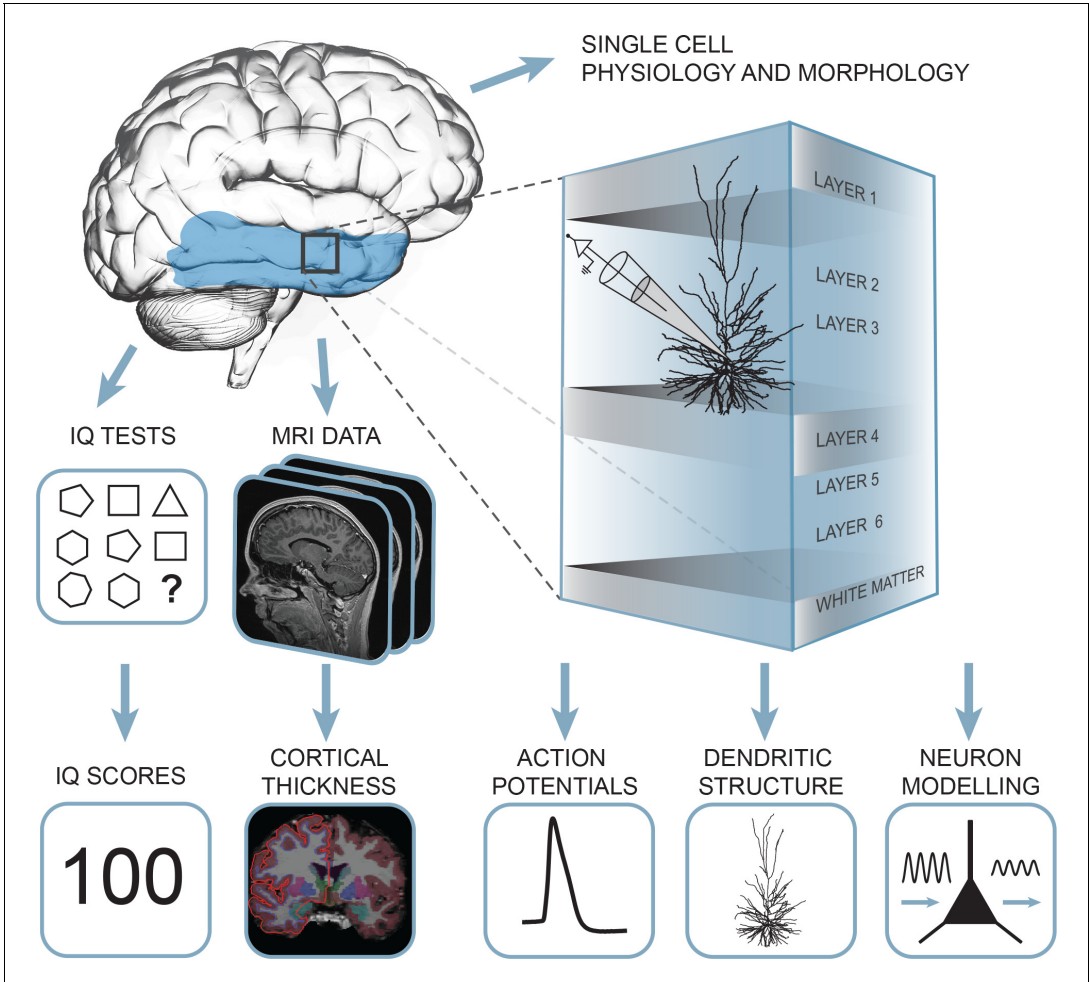

**Figure 1.** Summary of the approach: multidimensional data set from human subjects contained single cell physiology, neuronal morphology, MRI and IQ test scores (WAIS FSIQ). The area of the brain highlighted in blue indicates the location of cortical thickness measurements, black square indicates the typical origin of resected cortical tissue.

DOI: https://doi.org/10.7554/eLife.41714.003

The following figure supplements are available for figure 1:

**Figure supplement 1.** Subject disease history and age do not correlate with IQ and neuronal morphology.
DOI: https://doi.org/10.7554/eLife.41714.004

**Figure supplement 2.** Neuronal morphology, IQ or AP rise speed are not different across patient groups.
DOI: https://doi.org/10.7554/eLife.41714.005

posterior to the temporal pole (*Figure 2b*) as a block of ~1–1.5 cm in diameter and was removed to gain access to the disease focus in deeper lying structures such as hippocampus or amygdala. In all patients, the resected neocortical tissue was not part of the epileptic focus or tumor and displayed no structural/functional abnormalities in preoperative MRI investigation, electrophysiological whole-cell recordings or microscopic investigation of histochemically stained tissue (*Mohan et al., 2015*; *Testa-Silva et al., 2014*; *Testa-Silva et al., 2010*; *Verhoog et al., 2016*; *Verhoog et al., 2013*). In line with the non-pathological status of tissue, we observed no correlations of cellular parameters or IQ scores with the subject's disease history and age (*Figure 1—figure supplements 1–2*). After resection the tissue was immediately placed in ice-cold artificial cerebro-spinal fluid (aCSF) and within 15 min transported to the lab, sliced and maintained to enable single cell physiological recordings and biocytin filling.

We recorded action potentials (APs) from human pyramidal neurons in superficial layers of temporal cortex and digitally reconstructed their complete dendritic structures. We tested the

**Table 1.** Subject details.

| Patient number | IQ | Age | Diagnosis | Gender | Antiepileptic drugs |
|---|---|---|---|---|---|
| 1 | 88 | 41 | Tumor | M | CBZ |
| 2 | 78 | 21 | Other | F | LEV; VPA |
| 3 | 119 | 66 | Tumor | F | None |
| 4 | 88 | 31 | Tumor | F | CBZ; LEV |
| 5 | 81 | 51 | Other | F | CLB; LTG; OXC |
| 6 | 69 | 58 | MTS | F | CZP |
| 7 | 107 | 28 | Tumor | M | LTG; LEV |
| 8 | 115 | 29 | MTS | F | LTG; TPM |
| 9 | 125 | 20 | Tumor | M | CBZ; LEV |
| 10 | 84 | 27 | Tumor | F | CBZ, LTG |
| 11 | 110 | 41 | Tumor | M | CBZ; LTG |
| 12 | 87 | 18 | MTS | M | OXC |
| 13 | 67 | 23 | MTS | F | LEV; OXC |
| 14 | 72 | 53 | MTS | M | CBZ; CLB |
| 15 | 97 | 25 | Tumor | M | None |
| 16 | 104 | 19 | Other | M | CLB; OXC |
| 17 | 88 | 48 | Other | F | CBZ |
| 18 | 65 | 38 | MTS | F | CBZ; LEV |
| 19 | 62 | 40 | Other | F | None |
| 20 | 84.5 | 31 | Other | F | None |
| 21 | 88 | 35 | Other | F | CZP; LCS; LTG; LEV |
| 22 | 77 | 54 | Tumor | M | VPA |
| 23 | 91 | 25 | Other | M | CLB; LCS; LEV |
| 24 | 70 | 31 | MTS | F | CBZ; CLB |
| 25 | 114 | 49 | Other | M | CBZ; CLB; LEV |
| 26 | 83 | 25 | Tumor | M | None |
| 27 | 109 | 45 | Other | F | CBZ; CLB; LTG |
| 28 | 102 | 47 | Tumor | F | CBZ |
| 29 | 67 | 22 | Other | M | CLB; LTG; LEV |
| 30 | 97 | 38 | MTS | M | CBZ |
| 31 | 79 | 40 | MTS | F | CBZ, CLB, LTG, LEV |
| 32 | 117 | 44 | Other | M | LCS; VPA |
| 33 | 99 | 30 | Tumor | F | CLB; OXC |
| 34 | 72 | 44 | MTS | M | LTG; LEV |
| 35 | 82 | 41 | Other | F | CBZ, LEV, TPM |
| 36 | 95 | 29 | Other | M | CBZ; PB |
| 37 | 91 | 20 | Other | F | CBZ; LEV |
| 38 | 82 | 21 | Tumor | M | CBZ; LCS; LTG; LEV |
| 39 | 115 | 40 | MTS | M | CBZ; LEV |
| 40 | 97 | 48 | MTS | F | CBZ; ZNS |
| 41 | 94 | 40 | MTS | F | CLB; LTG; ZNS |
| 42 | 81 | 44 | MTS | M | CBZ; LTG |
| 43 | 70 | 33 | MTS | F | CBZ; CLB; LEV |
| 44 | 82 | 51 | Other | M | CBZ |

*Table 1 continued on next page*

Table 1 continued

| Patient number | IQ | Age | Diagnosis | Gender | Antiepileptic drugs |
|---|---|---|---|---|---|
| 45 | 114 | 18 | Tumor | F | OXC |
| 46 | 90 | 23 | Other | M | OXC |

M = male; F = female;

Antiepileptic drugs specified: Carbamazepine (CBZ); Lamotrigine (LTG); Levetiracetam (LEV); Topiramate (TPM); Clobazam (CLB); Oxcarbazepine (OXC); Clonazepam (CZP); Phenobarbital (PB); Phenytoin (PHT); Lacosamide (LCS); Sodium valproate (VPA); Zonisamide (ZNS)

DOI: https://doi.org/10.7554/eLife.41714.007

hypothesis that variation in neuronal morphology can lead to functional differences in AP speed and information transfer and explain variation in IQ scores. In addition to our experimental results, we used computational modelling to understand underlying principles of efficient information transfer in human cortical neurons.

## Results

### IQ scores positively correlate with cortical thickness of the temporal lobe

Cortical thickness of the temporal lobe has been associated with IQ scores in hundreds of healthy subjects (*Choi et al., 2008*; *Deary et al., 2010*; *Hulshoff Pol et al., 2006*; *Karama et al., 2009*; *Narr et al., 2007*), and we first asked whether this applies to the subjects in our study as well. From T1-weighted MRI scans obtained prior to surgery, we determined temporal cortical thickness in 35 subjects using voxel-based morphometry of temporal cortical areas. These areas included the surgically resected cortical tissue (*Figure 2b*) used for cellular recordings and neuronal reconstructions, which typically came from locations at 4 cm from temporal pole and was 1–1.5 cm in diameter (black circle in *Figure 2b*). The total resected cortical area varied for each patient, but consisted of a larger part of the temporal lobe (*Figure 2b*; average resected area in red, maximum in orange). The mean distance of resection boundaries from temporal pole was 4.2 ± 1.7 cm on superior temporal gyrus, 4.8 ± 1.5 cm on middle temporal gyrus, and 4.9 ± 1.5 cm on inferior temporal gyrus for the 46 subjects in this study. In MRI images, cortical thickness was measured in temporal lobe that included the resection areas and corresponded to the areas identified to associate with IQ in healthy subjects (*Choi et al., 2008*; *Deary et al., 2010*; *Hulshoff Pol et al., 2006*; *Karama et al., 2009*; *Narr et al., 2007*) (*Figure 2c*; in red). The superior temporal gyrus was excluded from this analysis as it contains areas for auditory, gustatory and language processing that are spared during resection. Cortical thickness measurements were collapsed to one mean value for cortical thickness for each subject. In line with findings in healthy subjects (*Choi et al., 2008*; *Deary et al., 2010*; *Hulshoff Pol et al., 2006*; *Narr et al., 2007*; *Karama et al., 2009*) mean cortical thickness in temporal lobes positively correlated with IQ scores of the subjects (*Figure 2d*).

### IQ scores positively correlate with dendritic structure of temporal cortical pyramidal neurons

Cortical association areas in temporal lobes play a key role in high-level integrative neuronal processes and its superficial layers harbor neurons of increased neuronal complexity (*DeFelipe et al., 2002*; *Elston, 2003*; *Scholtens et al., 2014*; *van den Heuvel et al., 2015*). In rodents, the neuropil of cortical association areas consists for over 30% of dendritic structures (*Ikari and Hayashi, 1981*). To test the hypothesis that human temporal cortical thickness is associated with dendrite size, we used 72 full reconstructions of biocytin-labelled temporal cortical pyramidal neurons from layers 2, 3 and 4 (median number of neurons per subject = 2; average 2.8; ranging from 1 to 10) part of which was previously reported (*Mohan et al., 2015*). We calculated total dendritic length (TDL) that included all basal and apical dendrites without apparent slice artifacts for each neuron. We computed TDL from multiple neurons for each subject and correlated these mean TDL values to mean temporal cortical thickness from the same subject. We found that dendritic length positively

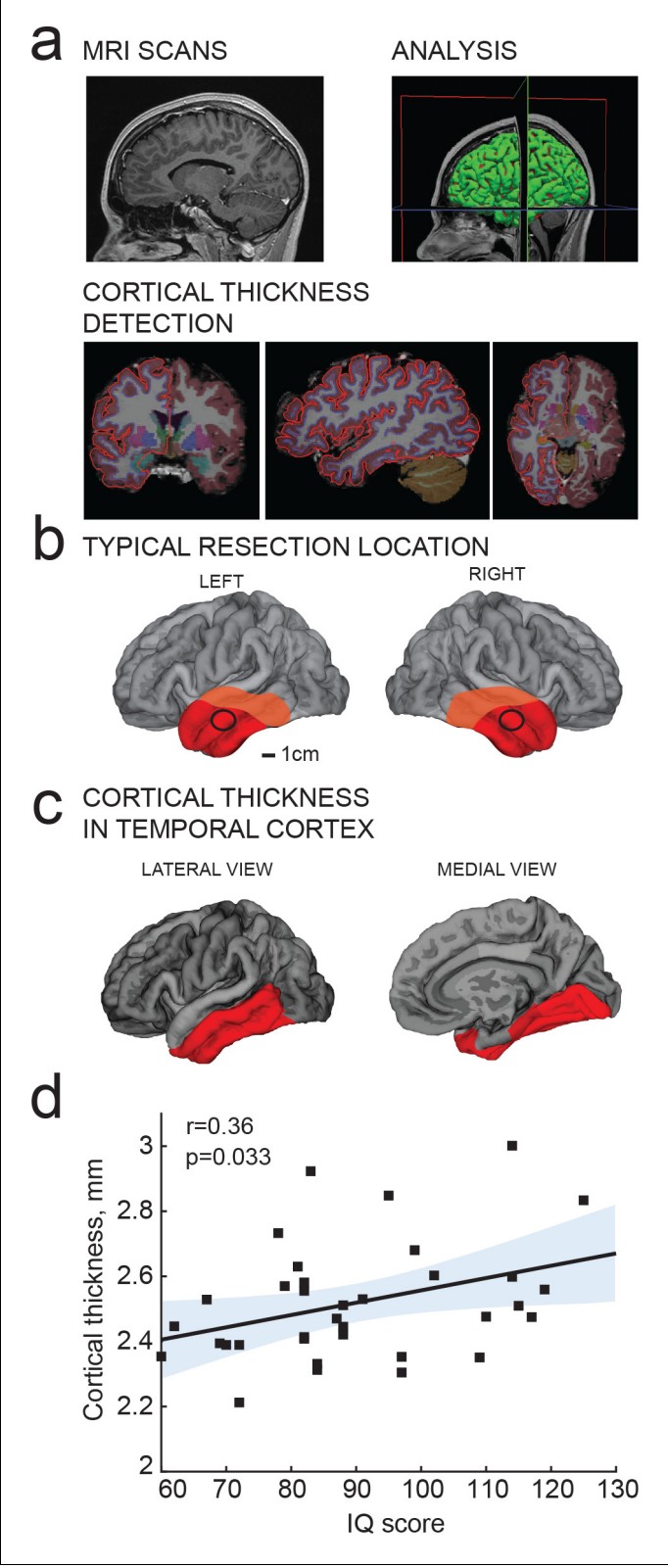

**Figure 2.** IQ scores positively correlate with cortical thickness of the temporal lobe. (**a**) MRI analysis pipeline: (1) Presurgical MRI T1-weighted scans; (2) Morphometric analysis; (3) Detection of cortical thickness from pial and white-grey matter boundaries; (**b**) Typical resection location for tissue used in this study is marked by a black circle; average total resected area from the patient is shown in red and maximum resected area in orange; (**c**) selection
*Figure 2 continued on next page*

*Figure 2 continued*

of temporal cortical area for correlations with IQ in b (red). (d) Average cortical thickness in temporal lobe (from area highlighted in red in c) positively correlates with IQ scores from the same subjects (n subjects = 35). Here and in figures below, Pearson correlation coefficients and p-values are reported in graph insets, the solid line represents linear regression ($R^2$ = 0.13), shaded area indicates 95% confidence bounds of the fit.

DOI: https://doi.org/10.7554/eLife.41714.006

correlated with mean temporal lobe cortical thickness (Pearson correlation coefficient r = 0.5, explained variance $R^2$ = 0.25), indicating that dendritic structure of individual neurons contributes to the overall cytoarchitecture of temporal cortex (*Figure 3a*).

TDL is in part determined by the soma location within cortical layers: cell bodies of pyramidal neurons with larger dendrites typically lie deeper, at larger distance from pia (*Mohan et al., 2015*). To exclude a systematic bias in sampling, we determined the cortical depth of each neuron relative to the subject's temporal cortical thickness in the same hemisphere. There was no correlation between IQ score and relative cortical depth of pyramidal neurons indicating that we sampled neurons at similar depths across subjects (*Figure 3b*). Finally, we tested whether mean TDL and complexity of pyramidal neurons relates to subjects' IQ scores. We found a strong positive correlation between individual's pyramidal neuron TDL and IQ scores (Pearson correlation coefficient r = 0.51, explained variance $R^2$ = 0.26; *Figure 3c*) as well as between number of dendritic branch points and IQ scores (r = 0.46, $R^2$ = 0.22; *Figure 3d*). Thus, larger and more complex pyramidal neurons in temporal association area may partly contribute to thicker cortex and link to higher intelligence.

## Larger dendrites lead to faster AP onset and improved encoding properties

Dendrites not only receive most synapses in neurons, but dendritic morphology and ionic conductances act in concert to regulate neuronal excitability (*Bekkers and Häusser, 2007*; *Eyal et al., 2014*; *Vetter et al., 2001*). In model simulations where neurons are reduced to balls and sticks, increasing the dendritic membrane surface area, that is the dendritic impedance load, speeds up the onset phase of APs. This is a consequence of the decrease in the effective time constants of the neuron with increasing dendritic size and dendritic impedance load (*Eyal et al., 2014*). Larger dendrites act as a larger sink for currents generated in the axon initial segment during AP onset and result in faster membrane potential changes. Furthermore, we found previously that human neocortical pyramidal neurons, which are three times larger than rodent pyramidal neurons (*Mohan et al., 2015*), have faster AP onsets compared to rodent neurons and are able to track and encode fast synaptic inputs and sub-threshold changes in membrane potential with high temporal precision (*Testa-Silva et al., 2014*). We asked whether the observed differences in TDL between human pyramidal neurons affected their encoding properties and ability to transfer information. To this end, we incorporated the 3-dimensional dendritic reconstructions of the 72 human pyramidal neurons into in silico models, equipped them with excitable properties (see Materials and methods) and tested whether their APs have faster onset. We found that TDL of model neurons with realistic dendritic trees positively correlated with the steepness of AP onsets (r = 0.4, $R^2$ = 0.16; *Figure 4a,b*) and larger dendrites enabled neurons to generate faster APs.

The exact timing of action potential firing allows cortical neurons to pass on temporal information provided by synaptic inputs (*Köndgen et al., 2008*; *Ilin et al., 2013*; *Testa-Silva et al., 2014*; *Linaro et al., 2018*). Single pyramidal neurons do not sustain high frequency firing and generally do not encode high frequency synaptic input content in rate coding. Instead, the precision in timing of AP initiation does allow these neurons to encode incoming high frequency information in their output. In contrast to rodent neurons, human neurons can encode sub-threshold membrane potential changes on a sub-millisecond timescale by timing of APs (*Testa-Silva et al., 2014*). This synaptic input tracking capacity strongly relies on the rapidity of AP onset (*Ilin et al., 2013*). Faster APs allow neurons to respond to fast synaptic inputs, which will be missed if AP generation is too slow. Thereby, neurons with faster APs can translate higher frequencies of synaptic membrane potential fluctuations into AP timing and ultimately encode more information.

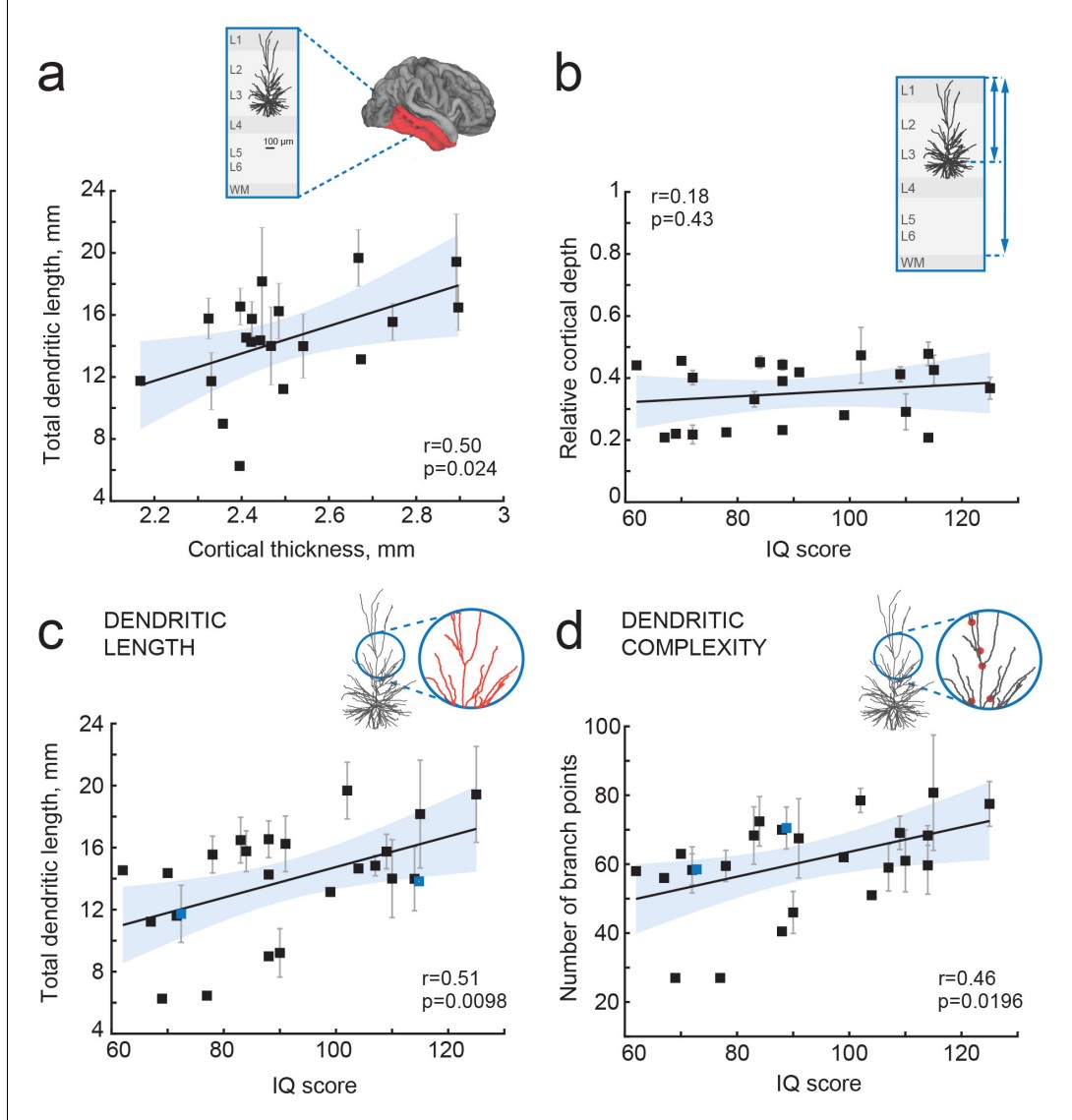

**Figure 3.** IQ scores positively correlate with dendritic structure of temporal cortical pyramidal cells. (a) Average total dendritic length in pyramidal cells in superficial layers of temporal cortex positively correlates with cortical thickness in temporal lobe from the same hemisphere (area shaded in a, n subjects = 20; n neurons = 57, $R^2$ = 0.25). Inset shows a scheme of cortical tissue with a digitally reconstructed neuron and the brain area for cortical thickness estimation (red) (b) Cortical depth of pyramidal neurons, relative to cortical thickness in temporal cortex from the same hemisphere, does not correlate with IQ score (n subjects = 21, $R^2$ = 0.03). Inset represents the cortical tissue, blue lines indicate the depth of neuron and cortical thickness (c) Total dendritic length (TDL) and (d) number of dendritic branches positively correlate with IQ scores from the same individuals (n subjects = 25, n neurons = 72, TDL $R^2$ = 0.26, Branch points $R^2$ = 0.22). Symbols highlighted in blue were shifted along the x axis for display purposes. Data are mean per subject ±standard deviation.

DOI: https://doi.org/10.7554/eLife.41714.008

The aforementioned theoretical work (*Eyal et al., 2014*) using 'ball-and-stick' neuron models showed that neurons with larger dendritic compartments not only have faster AP onset rapidity, but could also time AP generation to faster changes in membrane potentials, increasing the frequency tracking capabilities of input modulations, and augmenting the input frequency bandwidth of information encoding about three times. However, it is not known whether the same effect holds true for the human cortical pyramidal neurons we recorded from, and whether the range of dendritic compartment sizes we examined might lead to significant quantitative biophysical differences. We tested this by simulating sinusoidal current inputs of increasing frequencies into in silico representations of

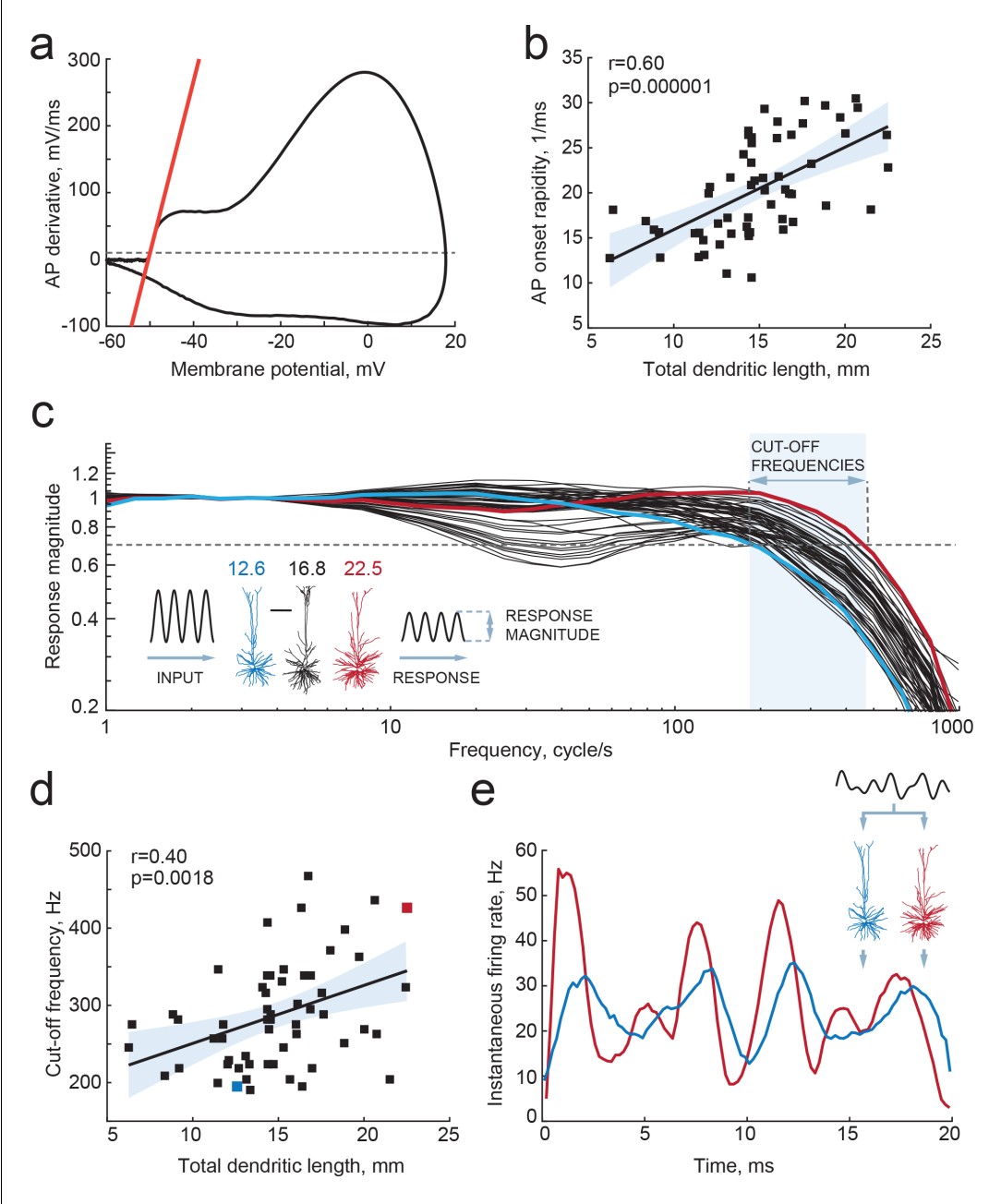

**Figure 4.** Larger dendrites lead to faster AP onset and improved encoding properties. (a,b) Higher TDL results in faster onsets of model-generated APs: (a) example phase plot of an AP is shown with a red line representing onset rapidity - slope of AP derivative at 10 mV/ms (grey dashed line); (b) onset rapidity values of simulated APs positively correlate with TDL ($R^2$ = 0.36). (c) Model neurons received simulated sinusoidal current-clamp inputs and generated spiking responses of different magnitudes and frequencies. Red and blue traces are response magnitudes of example neurons with low (blue) and large (red) TDLs; inset shows examples of morphological reconstructions with their TDLs in mm shown above. Cut-off frequencies are defined within the frequency range (shaded area) at which the model neuron can still track the inputs reliably (produce response of 0.7 response magnitude, dashed line). (d) Cut-off frequencies positively correlate with TDL ($R^2$ = 0.16; example neurons from panel (c) are highlighted by the same colors). (e) Responses to the same input in two example neurons from panels (b) and (c): instantaneous firing frequency of the model neuron with large TDL (red) follows the input with higher temporal precision than the model neuron with smaller TDL (blue).
DOI: https://doi.org/10.7554/eLife.41714.009

the neurons we recorded and reconstructed, and studied how the timing of AP firing of these neurons followed sub-threshold membrane potential changes. We find that human neurons with larger TDL can reliably time their APs to faster membrane potential changes, with cut-off frequencies up to 400–500 Hz, while smaller neurons had their cut-off frequencies already at 200 Hz (*Figure 4c,d*). Furthermore, there was a significant positive correlation between the dendritic length and the cut-off frequency (*Figure 4d*). Finally, given the same input - composed of the sum of three sinusoids of increasing frequencies - larger neurons were able to better encode rapidly changing temporal information into timing of AP firing, compared to smaller neurons (*Figure 4e*). Thus, we find that differences in dendritic length of human neurons lead to faster APs and thereby to wider frequency bandwidths of encoding synaptic inputs into timing of AP output.

## Higher IQ scores associate with faster APs

Since cortical pyramidal neurons with large dendrites have faster APs and can encode more information in AP output, and since large dendrites also associate with higher IQ scores, we next asked whether human cortical pyramidal neurons from individuals with higher IQ scores generate faster APs. To test this, we made whole-cell recordings from pyramidal cells in acute slices of temporal cortex (31 subjects, 129 neurons, median number of neurons per subject = 3; ranging from 1 to 11 *Figure 5*) and recorded APs at different firing frequencies in response to depolarizing current steps. We determined AP maximum rise speed, which is highly correlated with AP onset rapidity (r = 0.79 p=4.29e-14, n = 60, data not shown), and can more reliably be determined from recordings with sampling frequencies between 10 and 50 kHz. Maximum rise speed of APs depended on the firing history of the cell, with the first AP in the train having the highest AP rise speed and slowing down with increasing instantaneous firing frequency, the time interval between subsequent APs (*Figure 5b–d*). To test whether AP rise speed differed between IQ groups, we split all AP rise speed data into two groups based on IQ score – above and below 100. Although the AP rise speed of the first AP was not different between high and low IQ groups (*Figure 5c*), the AP slowed down stronger in individuals with lower IQ scores compared to APs of individuals with higher IQ scores (*Figure 5d*). At higher instantaneous firing frequencies (20–40 Hz), the AP rise speed was higher in individuals with IQ scores above 100 (*Figure 5c* right; AP rise speed high IQ = 338.4 ± 26.03 mV/ms; AP rise speed low IQ = 268.1 ± 12.20 mV/ms, t-test p=0.0113). We next calculated the slowing of APs with increasing instantaneous frequency by normalizing rise speed of APs to the rise speed of the first AP in the train. Relative to first AP, rise speed at 20–40 Hz showed significant slowing in subjects with lower IQ scores and decreased to 74% of the initial AP rise speed. In contrast, in neurons from individuals with higher IQ scores, AP rise speed remained on average at 84% (*Figure 5d* right, high IQ = 0.84 ± 0.014; low IQ = 0.74 ± 0.024, t-test p value=0.037).

We further investigated whether these differences at the group level reflected correlations between individual IQ scores and AP rise speeds. We correlated mean AP rise speeds both of the first AP and AP at 20–40 Hz from all neurons of the same subject to the subject's IQ score. The AP rise speed of the first AP in the train positively correlated with IQ scores (r = 0.41, $R^2$ = 0.17; *Figure 5e*), and this correlation was even stronger for AP rise speeds at instantaneous frequencies of 20–40 Hz (r = 0.46, $R^2$ = 0.21; *Figure 5f*). Importantly, also relative AP values showed significant positive correlations with IQ, indicating that it is the relative slowing of APs at high frequencies that associates with intelligence (r = 0.37, $R^2$ = 0.14; *Figure 5g*). Finally, we asked whether the slowing of APs relates to the dendritic size of the same neurons, as our model results suggest. We find that larger neurons show less slowing of AP rise speed (higher relative AP speeds) at 20–40 Hz (r = 0.55, $R^2$ = 0.30; *Figure 5h*). These findings reveal that higher IQ scores are accompanied by faster APs during repeated AP firing, while lower IQ scores associate with increased AP fatigue during elevated neuronal activity. Thus, neurons from individuals with higher IQ scores are better equipped to process synaptic signals at high rates and at faster time scales, which is necessary to encode large amounts of information accurately and efficiently.

## Discussion

Our findings provide a first insight into the possible cellular nature of human intelligence and explain individual variation in IQ scores based on neuronal properties: faster AP rise speed during neuronal activity and more complex, extended dendrites associate with higher intelligence. AP kinetics have

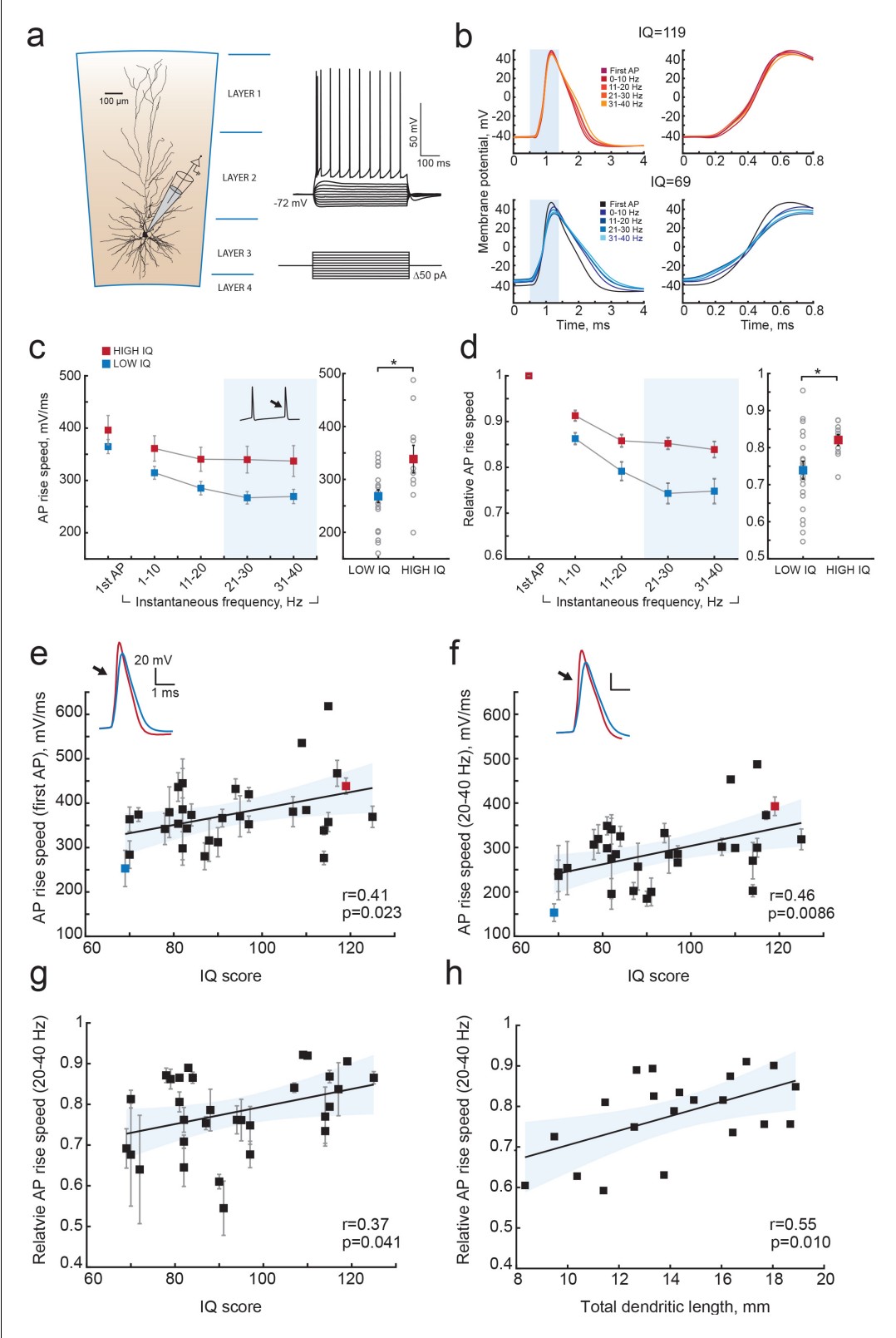

**Figure 5.** Higher IQ scores associate with faster AP initiation. (**a**) Scheme of a whole-cell recording showing biocytin reconstruction of a pyramidal neuron from human temporal cortex. Right: typical voltage responses to depolarizing somatic current injections. (**b**) Examples of AP traces at increasing instantaneous firing frequencies (frequency is shown in color code in insets) recorded from a subject with IQ = 119 (above panel, red) and a subject with IQ = 69 (lower panel, blue). AP rising phase in shaded area is displayed to the right (**c**) APs from subjects with higher IQ are better able to maintain

*Figure 5 continued on next page*

*Figure 5 continued*

their rise speed at increasing frequencies. Average (per neuron and subject) AP rise speed and (**d**) relative to first AP rise speeds in neurons from subjects with IQ < 100 (red, n subjects = 21, n neurons = 91) and subjects with IQ > 100 (blue, n subjects = 10, n neurons = 38) are displayed against instantaneous firing frequency. Right: data points in shaded area are shown as averaged values for 20–40 Hz (filled squares are group means, open circles are mean rise speeds per subject), *p<0.05. (**e**) IQ scores positively correlate with the rise speeds of first AP in the train (n subjects 31, n neurons = 129; $R^2$ = 0.17), (**f**) AP rise speed at 20–40 Hz (same data as right panel in (**c**), $R^2$ = 0.21) and (**g**) relative AP rise speeds at 20–40 Hz (same data as right panel in (**d**), $R^2$ = 0.14). (**h**) Larger neurons show less slowing of AP rise speed at higher frequencies: relative AP rise speeds at 20–40 Hz for individual neurons are plotted as a function of their TDL (n = 21 neurons, $R^2$ = 0.30). In c,d data are mean per subject ±S.E.M; in e, f, g data are mean ±standard deviation.

DOI: https://doi.org/10.7554/eLife.41714.010

profound consequences for information processing. In vivo, neurons are constantly bombarded by high frequency synaptic inputs and the capacity of neurons to keep track and phase-lock to these inputs determines how much of this synaptic information can be passed on to other neurons (*Testa-Silva et al., 2014*). The brain operates at a millisecond time-scale and even sub-millisecond details of spike trains contain behaviorally relevant information that can steer behavioral responses (*Nemenman et al., 2008*). Indeed, one of the most robust and replicable findings in behavioral psychology is the association of intelligence scores with measures of cognitive information-processing speed (*Barrett et al., 1986*). Specifically, reaction times (RT) in simple RT tasks provide a better prediction of IQ than other speed-of-processing tests, with a regression coefficient of 0.447 (*Vernon, 1983*). In addition, high positive correlations between RT and other speed-of-processing tests suggest the existence of a common mental speed factor (*Vernon, 1983*). Recently, these classic findings were confirmed in a large longitudinal population-based study counting more than 2000 participants (*Der and Deary, 2017*). Especially strong correlations between RT and general intelligence were reported for a slightly more complex 4-choice (*Der and Deary, 2017*). Our results provide a biological cellular explanation for such mental speed factors: in conditions of increased mental activity or more demanding cognitive task, neurons of individuals with higher IQ are able to sustain fast action potentials and can transfer more information content from synaptic input to AP output.

Pyramidal cells are integrators and accumulators of synaptic information. Larger dendrites can physically contain more synaptic contacts and integrate more information. Indeed, human pyramidal neuron dendrites receive twice as many synapses as in rodents (*DeFelipe et al., 2002*) and cortico-cortical whole-brain connectivity positively correlates with the size of dendrites in these cells (*Scholtens et al., 2014*; *van den Heuvel et al., 2015*). In this and a previous study (*Mohan et al., 2015*), we find almost 2-fold larger dendritic arbor size (mean TDL = 14.67 ± 4 mm) and number of dendritic branches (64.03 ± 17.7) compared to reports that use post-mortem tissue (*Jacobs et al., 2001*; *Bianchi et al., 2013*; *Elston et al., 2001*). The differences could be explained by a number of advantages of biocytin filled neurons in surgical resections compared to traditionally used Golgi stainings in human post-mortem tissue. The cortical slices in our study are thicker (350 μm compared to 120–250 μm) and contain neurons with almost completely intact apical and basal dendrites, while other studies use only basal dendrites for quantification (*Jacobs et al., 2001*). Furthermore, only a small number of neurons are filled in a slice, which allows to unambiguously quantify all dendrites from individual cells. Importantly, the tissue comes from a living donor compared to post-mortem tissue collection, and thus does not suffer from post-mortem delays (*de Ruiter, 1983*) and only still living functional cells are filled. At the same time, post-mortem studies make it possible to make comparative analysis of several cortical areas. A gradient in complexity of pyramidal cells in cortical superficial layers accompanies the increasing integration capacity of cortical areas, indicating that larger dendrites are required for higher-order cortical processing (*Elston et al., 2001*; *Jacobs et al., 2001*; *van den Heuvel et al., 2015*). Our results align well with these findings, suggesting that the neuronal complexity gradient also exists from individual to individual and could explain differences in mental ability.

Within human cortex, association areas contain neurons with larger and more complex dendrites than primary sensory areas, while neuronal cell body density is lower in cortical association areas compared to primary sensory areas (*Elston, 2003*; *DeFelipe et al., 2002*). Larger neurons are not as tightly packed together within cortical space as smaller cells. A recent study by *Genç et al., 2018* used multi-shell diffusion tensor imaging to estimate parieto-frontal cortical dendritic density and

found that higher IQ scores correlated with lower values of dendritic density (*Genç et al., 2018*). This may indicate that parieto-frontal cortical areas in individuals with higher IQ scores have less densely packed neurons, and may suggest that these neurons are larger. In our study, we carefully determined the amount and complexity of dendrite for each neuron, a computational unit within the cortex with well-defined input-output signals. Taking the results of *Genç et al., 2018* and our study together may suggest that the neuronal circuitry associated with higher intelligence is organized in a sparse and efficient manner, where larger and more complex pyramidal cells occupy larger cortical volume.

Larger dendrites have an impact on excitability of cells (*Bekkers and Häusser, 2007*; *Vetter et al., 2001*) and determine the shape and rapidity of APs (*Eyal et al., 2014*). Increasing the size of dendritic compartments in silico lead to acceleration of AP onset and increased encoding capability of neurons (*Eyal et al., 2014*). Both in models and in slice recordings, changes of AP initiation dynamics were shown to fundamentally modify encoding of fast changing signals and the speed of communication between ensembles of cortical neurons (*Eyal et al., 2014*; *Ilin et al., 2013*). Neurons with fast AP onsets can encode high frequencies and respond quickly to subtle input changes. This ability can be impaired and response speed is decreased when AP onsets are slowed down by experimental manipulations (*Ilin et al., 2013*). Our results not only demonstrate that AP speed depends on dendritic length and influences information transfer, but also show that both dendritic length and AP speed in human neurons correlate with intelligence. Thus, individuals with larger dendrites are better equipped to transfer synaptic information at higher frequencies.

Remarkably, dendritic morphology and different parameters of AP waveform are also parameters that we have previously identified as showing pronounced differences between humans and other species (*Mohan et al., 2015*; *Testa-Silva et al., 2014*). Human pyramidal cells in layers 2 and 3 have 3-fold larger and more complex dendrites than in macaque or mouse (*Mohan et al., 2015*). Moreover, human APs have lower firing threshold and faster AP onset kinetics both in single APs and during repeated firing (*Testa-Silva et al., 2014*). These differences across species may suggest evolutionary pressure on both dendritic structure and AP waveform and emphasize specific adaptations of human pyramidal cells in association areas for cognitive functions.

Our results were obtained from patients undergoing neurosurgical procedure and, thus, may potentially raise questions on how representative our findings are for normal healthy human subjects. Although no healthy controls can be used for single cell measurements, we addressed this issue in the following way. Firstly, in all patients, the resected neocortical tissue was not part of epileptic focus or tumor and displayed no structural or functional abnormalities in preoperative MRI, electrophysiological recordings or microscopic investigation of stained tissue. Secondly, none of the parameters correlated with age at epilepsy onset, seizure frequency, age or disease duration (*Figure 1— figure supplement 1*). Thirdly, IQ, dendritic length or AP rise speed were not different across different patient groups (*Figure 1—figure supplement 2*). Finally, the cortical thickness correlation with general intelligence we observe in our study was also reported in hundreds of healthy subjects. Taken together, these results indicate that our findings are not likely to be influenced by disease background of the subjects.

In this study, intelligence was measured using WAIS IQ score, that combines results of 11 individual subtests of cognitive functioning into a single full-scale IQ score (*Wechsler, 2008*; *Taylor and Heaton, 2001*). This inevitably simplifies and reduces a multi-dimensional human trait to a single number. Although none of the intelligence tests can capture all aspects of human intelligence, IQ tests have proven their validity and relevance. The results of different cognitive subtests are highly correlated and generate a strong general factor – general intelligence or Spearman's *g* (*Spearman, 1904*). Spearman's *g*, calculated based on subtests of WAIS and expressed in total full-scale IQ score, strongly correlates with highly relevant life outcomes, including education, occupation, and income (*Strenze, 2007*; *Foverskov et al., 2017*). Moreover, intelligence is a stable trait over time in the same individual: the results of intelligence tests at the age of 11 predict the scores at the age of 90 (*Gow et al., 2011*; *Deary et al., 2013*). Thus, despite its shortcomings, full scale IQ score provides a relevant and meaningful estimation of general intelligence that lies at the core of cognitive differences between individuals.

In conclusion, our results provide first evidence that already at the level of individual neurons, such parameters as dendritic size and ability to maintain fast responses link to general mental ability. Multiplied by an astronomical number of cortical neurons in our brain, very small changes in these

parameters may lead to large differences in encoding capabilities and information transfer in cortical networks and result in a speed advantage in mental processing and, finally, in faster reaction times and higher cognitive ability.

## Materials and methods

### Human subjects and brain tissue

All procedures were performed with the approval of the Medical Ethical Committee of the VU University Medical Centre, and in accordance with Dutch license procedures and the Declaration of Helsinki. Written informed consent was provided by all subjects for data and tissue use for scientific research. All data were anonymized.

Human cortical brain tissue was removed as a part of surgical treatment of the subject in order to get access to a disease focus in deeper brain structures (hippocampus or amygdala) and typically originated from gyrus temporalis medium (Brodmann area 21). Speech areas were avoided during resection surgery through functional mapping. We obtained neocortical tissue from 46 patients (24 females, 22 males; age range 18–66 years, *Table 1*) treated for mesial temporal sclerosis, removal of a hippocampal tumor, low grade hippocampal lesion, cavernoma or another unspecified temporal lobe pathology. From 35 of these patients, we also obtained pre-surgical MRI scans, from 31 patients we recorded Action Potentials from 129 neurons and from 25 patients we had fully reconstructed dendritic morphologies from 72 neurons.

In all patients, the resected neocortical tissue was not part of epileptic focus or tumor and displayed no structural/functional abnormalities in preoperative MRI investigation, electrophysiological whole-cell recordings or microscopic investigation of stained tissue. The physiological recordings, subsequent morphological reconstructions, morphological and action potential analysis were performed blind to the IQ of the patients.

### IQ scores

Total IQ scores were obtained from all 46 subjects using the Dutch version of Wechsler Adult Intelligence Scale-III (WAIS-III) (*Taylor and Heaton, 2001*) and in some cases WAIS-IV (*Wechsler, 2008*) and consisted of following subtests: information, similarities, vocabulary, comprehension, block design, matrix reasoning, visual puzzles, picture comprehension, figure weights, digit span, arithmetic, symbol search and coding.

The tests were performed as a part of neuropsychological examination shortly before surgery, typically within one week.

### MRI data and cortical thickness estimation

T1-weighted brain images (1 mm thickness) were acquired with a 3T MR system (Signa HDXt, General Electric, Milwaukee, Wisconsin) as a part of pre-surgical assessment (number of slices = 170–180). Cortical reconstruction and volumetric segmentation was performed with the Freesurfer image analysis suite (http://freesurfer.net) (*Fischl and Dale, 2000*). The processing included motion correction and transformation to the Talairach frame. Calculation of the cortical thickness was done as the closest distance from the grey/white boundary to the grey/CSF boundary at each vertex and was based both on intensity and continuity information from the entire three-dimensional MR volume (*Fischl and Dale, 2000*). Neuroanatomical labels were automatically assigned to brain areas based on Destrieux cortical atlas parcellation as described in (*Fischl et al., 2004*). For averaging, the regions in temporal lobes were selected based on Destrieux cortical atlas parcellation in each subject.

### Slice preparation

Upon surgical resection, the cortical tissue block was immediately transferred to ice-cold artificial cerebral spinal fluid (aCSF) containing in (mM): 110 choline chloride, 26 NaHCO3, 10 D-glucose, 11.6 sodium ascorbate, 7 MgCl2, 3.1 sodium pyruvate, 2.5 KCl, 1.25 NaH2PO4, and 0.5 CaCl2 (300 mOsm) and transported to the neurophysiology laboratory (within 500 m from the operating room). The transition time between resection of the tissue and the start of preparing slices was less than 15 min. After removing the pia and identifying the pia-white matter axis, neocortical slices (350 μm

thickness) were prepared in ice-cold slicing solution (same composition as described above). Slices were then transferred to holding chambers in which they were stored for 30 min at 34 °C and for 30 min at room temperature before recording in aCSF, which contained (in mM): 125 NaCl; 3 KCl; 1.2 NaH2PO4; 1 MgSO4; 2 CaCl2; 26 NaHCO3; 10 D-glucose (300 mOsm), bubbled with carbogen gas (95% $O_2$/5% $CO_2$), as described previously (*Mohan et al., 2015*; *Testa-Silva et al., 2014*; *Testa-Silva et al., 2010*; *Verhoog et al., 2013*; *Verhoog et al., 2016*).

## Electrophysiological recordings

Cortical slices were visualized using infrared differential interference contrast (IR-DIC) microscopy. After the whole cell configuration was established, membrane potential responses to steps of current injection (step size 30–50 pA) were recorded. None of the neurons showed spontaneous epileptiform spiking activity. Recordings were made using Multiclamp 700A/B amplifiers (Axon Instruments) sampling at frequencies of 10 to 50 kHz, and lowpass filtered at 10 to 30 kHz. Recordings were digitized by pClamp software (Axon) and later analyzed off-line using custom-written Matlab scripts (MathWorks). Patch pipettes (3–5 MOhms) were pulled from standard-wall borosilicate capillaries and filled with intracellular solution containing (in mM): 110 K-gluconate; 10 KCl; 10 HEPES; 10 K-phosphocreatine; 4 ATP-Mg; 0.4 GTP, pH adjusted to 7.3 with KOH; 285–290 mOsm, 0.5 mg/ml biocytin. All experiments were performed at 32–35 °C. Only cells with bridge balance of <20 MOhm were used for further analysis.

## Morphological analysis

During electrophysiological recordings, cells were loaded with biocytin through the recording pipette. After the recordings the slices were fixed in 4% paraformaldehyde and the recorded cells were revealed with the chromogen 3,3-diaminobenzidine (DAB) tetrahydrochloride using the avidin–biotin–peroxidase method (*Horikawa and Armstrong, 1988*). Slices (350 μm thick) were mounted on slides and embedded in mowiol (Clariant GmbH, Frankfurt am Main, Germany). Neurons without apparent slicing artifacts and uniform biocytin signal were digitally reconstructed using Neurolucida software (Microbrightfield, Williston, VT, USA), using a × 100 oil objective. After reconstruction, morphologies were checked for accurate reconstruction in x/y/z planes, dendritic diameter, and continuity of dendrites. Finally, reconstructions were checked using an overlay in Adobe Illustrator between the Neurolucida reconstruction and Z-stack projection image from Surveyor Software (Chromaphor, Oberhausen, Germany). Only neurons with virtually complete dendritic structures were included; cells with major truncations due to slicing procedure were excluded.

Superficial layers pyramidal neurons were identified based on morphological and electrophysiological criteria at cortical depth within 400–1400 μm from cortical surface, that we previously found to correspond to cortical layers 2, 3 and 4 in humans (*Mohan et al., 2015*). For each neuron, we extracted total dendritic length (TDL) of all basal and apical dendrites and number of branch points and computed average TDL and average number of branch points for each subject by pooling data from all cells within one subject (1 to 10 neurons per subject). Only neurons without major truncations of apical dendrites by tissue sectioning were included for morphological analysis (*Mohan et al., 2015*; *Deitcher et al., 2017*).

## NEURON modelling

Following previous work (*Eyal et al., 2014*; *Eyal et al., 2016*) conductance-based multicompartmental 'Hodgkin and Huxley models' (*Hodgkin and Huxley, 1952*) of each of the reconstructed human pyramidal cells were built. To each model, a cylindrical axon (1 μm in diameter) was connected to the soma, consisting of a 50 μm long Axon Initial Segment (AIS) and a 1 mm long myelinated part. The AIS consisted of 25 electrical compartments, the rest of the axon of 21 compartments. Simulations were run with the open-source software simulator NEURON v.7.5 (*Carnevale and Hines, 2006*) (https://neuron.yale.edu/neuron), with dt = 10 μs integration time step at 37 °C. All compartments incorporated passive membrane properties, with specific capacitance $C_m$ = 0.75 μF/cm$^2$, axial resistance $R_a$ = 0.1 MOhm/cm, specific resistance $R_m$ = 30.3 MOhm/cm$^2$, and leak-currents with reversal potential E = −70 mV. In the myelinated part of the axon $C_m$ was decreased 37.5 times and $R_m$ was increased 5 times. Across all dendritic compartments, $C_m$ was increased by 84% and $R_m$ was decreased by the same amount to account for dendritic spines (*Sarid et al., 2007*; *Benavides-*

*Piccione et al., 2002*). Active membrane properties consisted of voltage-dependent fast-inactivating sodium (Na$^+$) and delayed-rectifier potassium (K$^+$) ionic conductances, taken from the SenseLab ModelDB database (*McDougal et al., 2017*) (https://senselab.med.yale.edu/modeldb) and set to: $g_{Na}$ = 0 pS/µm$^2$, $g_K$ = 0 pS/µm$^2$ in the myelinated axon, $g_{Na}$ = 8000 pS/µm$^2$ and $g_K$ = 1500 pS/µm$^2$ in AIS, $g_{Na}$ = 800 pS/µm$^2$ , $g_K$ = 320 pS/µm$^2$ in the soma, and $g_{Na}$ = 20 pS/µm$^2$ and $g_K$ = 10 pS/µm$^2$ for dendrites. Reversal potentials for Na$^+$ and K$^+$ currents were +50 mV and −85 mV, respectively. Resulting input resistances were 61.5±4.73 MOhm and resting potentials were −70.5±0.02 mV. Onset rapidity of simulated action potentials (APs) was calculated as the slope of membrane potentials *V(t)* in the phase plane (i.e. *V(t)* vs *dV/dt* at 10 mV/ms and averaged across APs in simulated trains.

The dynamical input-output 'transfer gain' (*Linaro et al., 2018*; *Köndgen et al., 2008*; *Testa-Silva et al., 2014*) was determined by injecting sinusoidally oscillating input currents for 120 s at the soma, with amplitude $I_1$, frequency $F$ (1–1'000 cycle/s), a DC baseline $I_0$ amplitude, and randomly fluctuating component $I_{noise}$:

$$I(t) = I_0 + I_1 sin(2\pi F t) + I_{noise}(t) \tag{1}$$

$I_{noise}(t)$ was an exponentially filtered stochastic Gaussian white-noise (Arsiero et al. 2007), with zero-mean, variance $s^2$ and correlation length $\tau_I$ = 5 ms, by iterating at each simulation time step:

$$I_{noise}(t+dt) = (1 - dt/\tau_I)I_{noise}(t) + s\sqrt{2dt/\tau_I}\xi_t \tag{2}$$

where $\{\xi_t\}$ is a sequence of independent pseudo-random Gaussian numbers. $s^2$ was set such that membrane potential hyperpolarization resulted in subthreshold potential fluctuations of ~3mV at -75 mV. DC baseline $I_0$ was set to induce mean firing rates of ~10 spike/s. $I_1$ was set to 1/6 of $I_0$.

AP firing times $\{t_k\}$ were detected at the soma and collected across all values of $F$. The output 'transfer gain'

$r_1(F)$ at a given frequency $F$ was defined as the amplitude of complex numbers in polar form:

$$r_1(F) = amplitude\left\{\sum_{j=1}^{N} exp(j2\pi Ft_k)\right\}/N \tag{3}$$

where N is the number of spikes and *j* is the imaginary unit. $r_1(F)$ was further normalized to $r_1(F_0)$, with $F_0 = 3$ cycle/s. The profile of $r_1(F)$ resembled a low-pass electrical filter, with cut-off frequency $F_c$ defined as the highest frequency at which $r_1(F_c) = r_1(F_0)/\sqrt{2}$. Input waveforms in *Figure 4*, inset, consisted of three rapidly varying components for 240 s:

$$I(t) = I_0 + I_1[sin(2\pi F_1 t) + sin(2\pi F_2 t) + sin(2\pi F_3 t)]/3 + I_{noise}(t) \tag{4}$$

with $F_1 = 200$, $F_2 = 300$, $F_3 = 450$ cycle/s.

## Action Potential waveform analysis of electrophysiological recordings

Action Potential (AP) waveforms were extracted from voltage traces recorded in response to intra-cellular current injections and sorted according to their instantaneous firing frequency. Instantaneous frequency was determined as 1/time to previous AP. Subsequently all APs were binned in 10 Hz bins, while the first APs in each trace were isolated in a separate bin.

AP rise speed was defined as the peak of AP derivative (dV/dt). For each analyzed cell, representative APs with all parameters were plotted for visual check to avoid errors in the analysis.

For each neuron, the mean values of AP rise speed in a given frequency bin were obtained by averaging all APs within that frequency bin. Relative AP rise speeds were calculated by dividing the mean AP rise speed in each frequency bin (1–10 Hz, 11–20 Hz, 21–30 Hz and 31 to 40 Hz) by the mean first AP rise speed (first APs in the train of APs).

To obtain AP values for each subject, AP parameters within each frequency bin were averaged for all neurons from one subject. All AP analysis was performed using customized Matlab scripts (source code available at https://github.com/INF-Rene/Morphys (*Verhoog et al., 2018*; copy archived at https://github.com/elifesciences-publications/Morphys).

## Statistical analysis

Statistical significance of all correlations between parameters was determined using Pearson correlation and linear regression (using Matlab, version R2017a, Mathworks). As multiple cells were measured per subject, correlations were calculated on mean parameter values per subject. All Pearson correlation coefficients and p values for correlations are shown in figure insets, $R^2$ coefficients and sample sizes are shown in figure legends and main text.

For statistical analysis of AP data, we divided all subjects according to their IQ into two groups: group with IQ > 100 and a group with IQ < 100. Differences between 2 IQ groups in AP rise times were statistically tested using Student t-test. For analysis of different patient groups (*Figure 1—figure supplements 2*) an ANOVA test was applied for each parameter separately.

## Acknowledgements

We thank Dr Linda Douw for her assistance with the analysis of brain imaging data and Mr. M Wijnants for his technical assistance and to the supercomputer facilities CalUA (University of Antwerp) for computing time. NAG received funding for this work from the from the Netherlands Organization for Scientific Research (NWO; VENI grant). HDM received funding for this work from the Netherlands Organization for Scientific Research (NWO; VICI grant), ERC StG 'BrainSignals', and EU H2020 'Human Brain Project' grant agreement no. 604102.

## Additional information

### Funding

| Funder | Grant reference number | Author |
| --- | --- | --- |
| Nederlandse Organisatie voor Wetenschappelijk Onderzoek | VENI grant | Natalia A Goriounova |
| H2020 European Research Council | 604102 | Michele Giugliano Huibert D Mansvelder |
| Fonds Wetenschappelijk Onderzoek | G0F1517N | Michele Giugliano |
| Nederlandse Organisatie voor Wetenschappelijk Onderzoek | VICI grant | Huibert D Mansvelder |
| H2020 European Research Council | ERC StG | Huibert D Mansvelder |

The funders had no role in study design, data collection and interpretation, or the decision to submit the work for publication.

### Author contributions

Natalia A Goriounova, Conceptualization, Formal analysis, Funding acquisition, Investigation, Visualization, Methodology, Writing—original draft, Writing—review and editing; Djai B Heyer, René Wilbers, Christophe Verbist, Software, Formal analysis; Matthijs B Verhoog, Software, Investigation; Michele Giugliano, Conceptualization, Software, Formal analysis, Methodology; Joshua Obermayer, Amber Kerkhofs, Harriët Smeding, Maaike Verberne, Sander Idema, Johannes C Baayen, Anton W Pieneman, Investigation; Christiaan PJ de Kock, Formal analysis, Investigation; Martin Klein, Conceptualization, Investigation; Huibert D Mansvelder, Conceptualization, Supervision, Funding acquisition, Writing—original draft, Project administration, Writing—review and editing

### Author ORCIDs

Natalia A Goriounova (iD) https://orcid.org/0000-0002-5917-983X
Michele Giugliano (iD) http://orcid.org/0000-0003-2626-594X
Huibert D Mansvelder (iD) https://orcid.org/0000-0003-1365-5340

## Ethics

Human subjects: All procedures were performed with the approval of the Medical Ethical Committee of the VU University Medical Centre (2012/362), and in accordance with Dutch license procedures and the Declaration of Helsinki. Written informed consent was provided by all subjects for data and tissue use for scientific research. All data were anonymized.

## Decision letter and Author response

Decision letter https://doi.org/10.7554/eLife.41714.016
Author response https://doi.org/10.7554/eLife.41714.017

## Additional files

### Supplementary files

• Transparent reporting form
DOI: https://doi.org/10.7554/eLife.41714.011

### Data availability

Numerical data for fall figures are available from the Dryad Digital Repository: https://doi.org/10.5061/dryad.83dv5j7. All customized Matlab scripts used for physiological data analysis are available at https://github.com/INF-Rene/Morphys (copy archived at https://github.com/elifesciences-publications/Morphys).

The following dataset was generated:

| Author(s) | Year | Dataset title | Dataset URL | Database and Identifier |
|---|---|---|---|---|
| Goriounova N, Heyer D, Wilbers R, Verhoog M, Giugliano M, Verbist C, Obermayer J, Kerkhofs A, Smeding H, Verberne M | 2018 | Data from: Large and fast human pyramidal neurons associate with intelligence | https://dx.doi.org/10.5061/dryad.83dv5j7 | Dryad Digital Repository, 10.5061/dryad.83dv5j7 |

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
