## [Decision Letter]

Thank you for submitting your article "Large and fast human pyramidal neurons associate with intelligence" for consideration by *eLife*. Your article has been reviewed by three peer reviewers, and the evaluation has been overseen by David Badre as Reviewing Editor and Timothy Behrens as the Senior Editor. The following individuals involved in review of your submission have agreed to reveal their identity: Christof Koch (Reviewer #1); John Duncan (Reviewer #2); Chet C Sherwood (Reviewer #3).

The reviewers have discussed the reviews with one another and the Reviewing Editor has drafted this decision to help you prepare a revised submission.

Summary:

This study tests the relationship between the structural and physiological properties of human temporal cortex pyramidal neurons and intelligence. The study analyzes a rare and unique dataset drawn from surgically resected, healthy temporal cortex samples. The prepared samples underwent morphological and electrophysiological tests and were related to individual measures of IQ. The results show that faster kinetics, enabled by larger dendritic trees, are related to higher IQ scores. This is a potentially exciting and impactful observation that connects levels of analysis from cells to circuits to behavior.

Essential revisions:

The comments from the reviewers for revision focused almost entirely on points that would enhance the clarity of the manuscript. Thus, as the reviewers have been clear on these points, I will relay them individually below:

- The neuronal modelling section of Materials and methods could do with some careful editing, as it is not written up to the standard of the remainder.

- IQ is obviously not a simple measure. Its validity, relationship to socioeconomic status and other factors has been much debated. I would recommend adding a short paragraph in the Introduction or Discussion to review possible limitations of IQ as an indicator of cognitive function.

- The boundaries for temporal cortex thickness measurement are not well described or justified. For example, it is evident in Figure 2A that superior temporal cortex is not included. Why not?

- It would be helpful to clarify how many of the 72 neurons that were traced came from each of the subjects. What was the median number traced from each subject, and range?

- More details on the location from the temporal lobe of resected tissue is important to know. How much variation was there in the location from patient to patient?

- For the measurement of TDL, it is not clear if this represents apical dendrites, basilar dendrites, or the entire dendritic tree. Also, it is said that only complete neurons were used in the analyses, yet it is not completely clear how thick the sections were that were used in morphological reconstructions. How do the current measurements of human temporal cortex pyramidal neurons compare to previous reports in the literature?

- To make it easier for the reader to follow, reviewer 3 suggests moving some of the important information about study design concerning the human patients and sources of the samples to the end of the Introduction or the beginning of the Results section. It is difficult to fully appreciate the basis of the Results without knowing the sources of the samples.

- In terms of completeness of citations, reviewer 3 was surprised that there were no references to the body of work from Bob Jacobs, who has published a large number of papers on the morphology of pyramidal neurons across the cortex in humans (normal variation, aging, and disease), as well as other species.

- Reviewer 3 suspects that the TLE effect on what they call "action potential onset rapidity" can be explained analytically by a simple ball-and-stick model of a RC circuit mimicking the soma and a sole dendrite of constant thickness and equivalent L (see the classical textbook Electric Current Flow by Jack, Noble and Tsien 1983 or my own Biophysics of Computation 1999). This would help sharpen the intuition of the reader.

Reviewer 1 had some additional questions regarding the results and data:

- Do the authors have enough data to compute what fraction of explained variance of their subject's IQ can be explained by their data?

- Combining their findings of Figure 3C (TDL increases with IQ) and Figure 4B (AP onset rapidity increases with TDL) leads me to conclude that AP onset rapidity will increase with IQ. How can they disentangle this expected relationship from the effect they report in Figure 5?

- Given the import of this study, the morphological, electrophysiological and suitably anonymized single cell data should be published.

---

## [Author Response]

Essential revisions:The comments from the reviewers for revision focused almost entirely on points that would enhance the clarity of the manuscript. Thus, as the reviewers have been clear on these points, I will relay them individually below:- The neuronal modelling section of Materials and methods could do with some careful editing, as it is not written up to the standard of the remainder.

We have altered the text of the Materials and methods modelling section to better fit the style of the section.

- IQ is obviously not a simple measure. Its validity, relationship to socioeconomic status and other factors has been much debated. I would recommend adding a short paragraph in the Introduction or Discussion to review possible limitations of IQ as an indicator of cognitive function.

A paragraph was added to the Discussion section to address the validity and limitations of IQ tests.

- The boundaries for temporal cortex thickness measurement are not well described or justified. For example, it is evident in Figure 2A that superior temporal cortex is not included. Why not?

A panel depicting the typical location of resection and the average and maximum resected areas has now been added to Figure 2B. In addition, we have added the average distances of resection boundaries to the text. For MRI analysis of cortical thickness, we selected the full area that was resected during surgery and that corresponded to the areas identified to associate with IQ in healthy subjects (Choi et al., 2008; Deary et al., 2010; Hulshoff Pol et al., 2006; Karama et al., 2009; Narr et al., 2007). The superior temporal gyrus was not included in this analysis, since it contains primary auditory and gustatory cortices and language processing areas that are spared from resection. This information is now included in the Results section (subsection “IQ scores positively correlate with cortical thickness of the temporal lobe”).

- It would be helpful to clarify how many of the 72 neurons that were traced came from each of the subjects. What was the median number traced from each subject, and range?

The median number of neurons per subject was 2 (average 2.8) and ranged from 1 to 10 neurons per subject. We added this information to the Results section text (subsection “IQ scores positively correlate with dendritic structure of temporal cortical pyramidal neurons”) as well as the information on the 129 neurons used for physiological analysis (median number of cells per subject = 3; ranging from 1 to 11) (subsection “Higher IQ scores associate with faster Aps”).

- More details on the location from the temporal lobe of resected tissue is important to know. How much variation was there in the location from patient to patient?

To address this comment, we added a panel depicting the typical location of resection, average and maximum resected areas to Figure 2B. Clarifying text describing the surgical procedure and mean and standard deviation of the resected area boundary from temporal lobe were added to the Results section text (subsection “IQ scores positively correlate with cortical thickness of the temporal lobe”).

- For the measurement of TDL, it is not clear if this represents apical dendrites, basilar dendrites, or the entire dendritic tree. Also, it is said that only complete neurons were used in the analyses, yet it is not completely clear how thick the sections were that were used in morphological reconstructions. How do the current measurements of human temporal cortex pyramidal neurons compare to previous reports in the literature?

In our study, TDL was calculated based on the full extent of both basal and apical dendrites, and only neurons were included of which apical dendrites were not truncated due to sectioning of the tissue. The thickness of sections was 350 µm. Clarifying text explaining this point was added to Results section (subsection “IQ scores positively correlate with dendritic structure of temporal cortical pyramidal neurons”) and Materials and methods section (subsection “Slice preparation”). To discuss the relation of our method to previous reports in the literature we added a paragraph in the Discussion section (second paragraph).

- To make it easier for the reader to follow, reviewer 3 suggests moving some of the important information about study design concerning the human patients and sources of the samples to the end of the Introduction or the beginning of the Results section. It is difficult to fully appreciate the basis of the Results without knowing the sources of the samples.

We added text to the Introduction section (third paragraph) to describe the source of the samples in more detail.

- In terms of completeness of citations, reviewer 3 was surprised that there were no references to the body of work from Bob Jacobs, who has published a large number of papers on the morphology of pyramidal neurons across the cortex in humans (normal variation, aging, and disease), as well as other species.

We regret omitting these highly relevant citations. We added a paragraph to the Discussion section to discuss our morphology results in the context of previously published reports including Bob Jacobs work (second paragraph).

- Reviewer 3 suspects that the TLE effect on what they call "action potential onset rapidity" can be explained analytically by a simple ball-and-stick model of a RC circuit mimicking the soma and a sole dendrite of constant thickness and equivalent L (see the classical textbook Electric Current Flow by Jack, Noble and Tsien 1983 or my own Biophysics of Computation 1999). This would help sharpen the intuition of the reader.

To help sharpen the intuition of the reader on this topic, we adjusted the text of the Results section (subsection “Larger dendrites lead to faster AP onset and improved encoding properties”, last paragraph), referring to results from ‘ball-and-stick’ modeling (Eyal et al., 2014), as suggested by the reviewer. In that study, the authors concluded that “increasing the dendritic membrane surface area (the dendritic impedance load) both enhances the AP onset in the axon and also shifts the cutoff frequency of the modulated membrane potential to higher frequencies. This combined “dendritic size effect” is the consequence of the decrease in the effective time constants of the neuron with increasing the dendritic impedance load (Eyal et al., 2014). In other words, larger dendrites act as larger sink for currents generated in the axon initial segment during AP onset and result in faster membrane potential changes.

Reviewer 1 had some additional questions regarding the results and data:- Do the authors have enough data to compute what fraction of explained variance of their subject's IQ can be explained by their data?

The explained variance – R^2^ coefficients – were mentioned in the figure legends for all correlations. For clarity, we now mention both r and R^2^ coefficients in the main text throughout the Results section.

- Combining their findings of Figure 3C (TDL increases with IQ) and Figure 4B (AP onset rapidity increases with TDL) leads me to conclude that AP onset rapidity will increase with IQ. How can they disentangle this expected relationship from the effect they report in Figure 5?

As the reviewer points out, the results of Figures 3 and 4 suggest that AP onset rapidity will increase with IQ, and in Figure 5 we present that AP max rise speed (as proxy of onset rapidity) indeed associates with IQ. The reviewer’s question is justified, are AP onset rapidity and AP max rise speed related? AP initiation occurs at a sub-millisecond time scale. Calculating AP onset rapidity of individual APs requires a sufficient number of data points, for which a minimum sampling rate of 50 kHz is required. In Figure 5, not all recordings met this requirement and some had sampling rates of 10 or 30 kHz. As a result, for several neurons from patients for which we had IQ scores, we could not reliably determine onset rapidity. However, 10 – 30 kHz is sufficient to calculate AP rise speed, which is calculated from the entire AP rise phase between threshold and peak. To address the reviewer’s concern, we calculated both parameters, AP onset rapidity and AP maximum rise speed, from the recordings that were done at a minimum of 50 kHz sampling rate (n=60) and tested whether these are associated. We found that onset rapidity and max rise speed of the same AP are strongly correlated with a Pearson correlation coefficient of 0.79 (see Author response image 1, in which the solid line represents linear regression, and the shaded area indicates 95% confidence bounds of the fit).

We added clarifying text and reported the r and p values for AP onset to AP rise speed correlations in the Results section (page 14, lines 322-324).

- Given the import of this study, the morphological, electrophysiological and suitably anonymized single cell data should be published.

Numerical data for all figures are available from the Dryad Digital Repository (doi number 10.5061/dryad.83dv5j7). All customized Matlab scripts used for physiological data analysis are available at https://github.com/INF-Rene/Morphys